# Prone Position Facilitates Creation of Ulnar-Basilic Arteriovenous Fistula

**DOI:** 10.3390/jcm11092610

**Published:** 2022-05-06

**Authors:** Tomasz Gołębiowski, Patryk Jerzak, Krzysztof Letachowicz, Andrzej Konieczny, Mariusz Kusztal, Maciej Gołębiowski, Mirosław Banasik, Katarzyna Sznajder, Magdalena Krajewska

**Affiliations:** 1Department of Nephrology and Transplantation Medicine, Wroclaw Medical University, Borowska 213, 50-556 Wroclaw, Poland; patryk_jerzak@o2.pl (P.J.); krzysztof.letachowicz@umw.edu.pl (K.L.); andrzej.konieczny@umw.edu.pl (A.K.); mariusz.kusztal@umw.edu.pl (M.K.); maciej.golebiowski@student.umw.edu.pl (M.G.); miroslaw.banasik@umw.edu.pl (M.B.); magdalena.krajewska@umw.edu.pl (M.K.); 2Department of Radiology, University of Opole, 45-052 Opole, Poland; katarzyna.sznajder@uni.opole.pl

**Keywords:** ulnar-basilicarteriovenous fistula, arteriovenous fistula, haemodialysis, end-stage renal disease

## Abstract

Background: The distal ulnar-basilic arteriovenous fistula (UBAVF) is a rarely used alternative type of vascular access for haemodialysis. The location of the vein on the back aspect of the forearm forces an extremely uncomfortable external rotation of the upper limb during surgery when the patient is in a supine position. Methods: We present a new approach towards creating UBAVF, which involves placing the patient in the prone position, thus eliminating the aforementioned inconvenience. The procedure was performed and described in a 46-year-old patient with chronic kidney disease (CKD) due to diabetic nephropathy. In the period from September 2021 to December 2021, we created an additional three UBAVFs with such modifications. Results: All fistulas were patent both immediately after the procedure and 2 weeks after surgery. Conclusions: The prone position may improve the comfort of both the operator and the patient during the procedure. On top of this, it may have a positive impact on the quality of the arteriovenous anastomosis.

## 1. Introduction

The National Kidney Foundation Kidney Disease Outcomes Quality Initiative (NKF-KDOQI) recommends the following order of haemodialysis access in patients expected to have a longer duration of haemodialysis (HD): forearm arteriovenous fistula (AVF) (snuffbox or distal radiocephalic (RCAVF) or transposed radiobasilic); forearm loop arteriovenous graft (AVG) or proximal forearm AVF (e.g., proximal radiocephalic, proximal vessel, and perforator combinations) or brachiocephalic; and brachiobasilic AVF or upper arm AVG [1].

The ulnar-basilic arteriovenous fistula (UBAVF) is a well-established type of vascular access, first described by Hanson et al. in 1967 [2]. In 1977, Kinnaert et al. reported a high rate of early failure and low survival of UBAVF in comparison with other techniques [3]. Similar results were obtained by Bourquelot et al. in both adults and children [4]. For this reason, distal UBAVF in the forearm is generally considered as a second or third choice of vascular access in both American [1] and European Vascular Access guidelines [5,6]. Inferior results compared to RCAVF may be explained by selection bias. UBAVF is attempted when the vessels are not suitable for creating RCAVF. Furthermore, the UBAVF procedure is more demanding due to the increased external rotation of the forearm required to maintain comfortable access to the vessels (Figure 1A). In case of patient fatigue, it is necessary to operate on the side of the forearm, which complicates the procedure. The prone position allows the tension-free positioning of the forearm and eliminates discomfort for both the operator and patient.

In our manuscript, we present a modified technique for creating UBAVF that has the potential to significantly improve the quality of arteriovenous anastomosis.

## 2. Case Description

The procedure was performed on a 46-year-old patient with chronic kidney disease (CKD) due to diabetic nephropathy following kidney transplantation (KTx) 18 years ago. The patient was admitted to our Clinic due to deterioration of the renal graft function and with a serum creatinine concentration of 5.74 mg/dL (range 0.8–1.3 mg/dL). Eventually, he was referred for haemodialysis access. Prior to KTx, a left RCAVF was used for haemodialysis; however, it thrombosed after the surgery. The initial ultrasound evaluation indicated the left proximal forearm as a potential AVF site. However, the procedure was unsuccessful, so the decision was made to create UBAVF in the distal forearm, i.e., the only feasible AVF site in the forearm.

## 3. Procedure Description

On the day preceding the surgery, an ultrasound assessment of the vessels of the left forearm was performed. The vessel diameters were as follows: radial artery—2 mm, ulnar artery—2.3 mm, brachial artery—4.6 mm, basilic vein, in the distal part of the forearm—3.9 mm, and this vein was patent along the entire length of the forearm. The result of the Allan test was negative. Next, the patient was placed in the prone position, and his tolerance was checked. He accepted this position and did not report any problems.

On the day of surgery, the patient was placed in the aforementioned position, and any ailments were controlled (Figure 1B). The procedure was performed under local anaesthesia using 1% lignocaine. An oblique incision was made in the distal part of the forearm. The venous and arterial vessels were dissected (Figure 2A), paying attention to avoid injury of the ulnar nerve. The artery showed advanced atherosclerotic changes. The vein was severed over the ligature, and the artery was cut longitudinally; afterwards, the two vessels, the end of the vein with the side of the artery, were anastomosed with a 6-0 insoluble polypropylene vascular suture. During the procedure, the operator used a magnifying glass with a 2.5-fold magnification. When the clamps were released, the vein filled with blood and a normal murmur was observed (Figure 2B). Finally, the wound was sutured with skin sutures. The fistula was patent both immediately and the day after the procedure, which was confirmed by ultrasound examination. The brachial blood flow was 330 mL/min. The patient was discharged with a functioning fistula. At the follow-up visit two weeks after the surgery, the fistula was still patent.

## 4. Discussion

The American [1] and European [5] guidelines provide the following order for the first three arteriovenous fistulas for HD: distal RCAVF, proximal forearm fistula and forearm loop graft. The significance of UBAVF is not precisely defined, although the aforementioned guidelines considered it as an alternative access to RCAVF.

The first report of UBAVF was published in 1967 [2], i.e., a year after the publication describing the Cimino–Brescia fistula [7]. Initial reports from the 1970s, provided by Klinneart et al. were not encouraging, as immediate thrombosis and two-year survival rates accounted for 20.7 and 60.9%, respectively [3]. In comparison, the two-year survival rate for RCAVF was 87.6%. The author also pointed out the high incidence of haematomas after the first dialysis due to the position on the back of the forearm and high vein motility. However, there are studies, including ours [8], that indicate much better results. Zehn et al. in a group of 44 patients with UBAVF and 321 with RCAVF, showed a similar patency rate. The primary patency rates after one and two years were 77.2% and 63.9% in the UBAVF group and 88.1% and 71.3% in the RCAVF group, respectively [9]. Similar conclusions were drawn by Sharma et al. [10], who compared the primary patency at 18 months and mean maturation time of 42 UBAVFs with 480 RCAVFs, which were 73.8%, 69.6% and 33.7 ± 6.6 days, 32.1 ± 4.7 days, respectively. Salgado et al. in 2004 reported 1-, 3-, and 5-year unassisted survival rates of 70.9%, 67.7%, and 57.3%, respectively, which were significantly higher than those documented previously. No episodes of surgical complications, arterial steal, or ulnar nerve damage were observed [11]. The authors recommended the inclusion of UBAVF in routine access plans. The microsurgical technique used in the creation of UBAVF by Bourquelot et al. resulted in immediate patency in 94% of 63 adults and 100% of seven children. However, only 60% of these fistulas were used as vascular access (38/63 adults and 6/7 children) after a mean postoperative interval of 80 days [12]. Undoubtedly, performing more procedures contributes to the experience of a center, and the additional use of new techniques—microscope [12] or loupe [10]—may improve the results of UBAVF creation and even approach the recent RCAVF outcomes.

RCAVF is a standard vascular access due to its simplicity and low complication rate. UBAVF, on the other hand, causes some technical difficulties related to a more difficult access to the basilic vein in relation to the cephalic vein, proximity of the ulnar nerve—the most important motor nerve of the forearm—longer maturation time and difficult cannulation to HD [10].

The main advantage of UBAVF is the low risk of steal syndrome and other complications associated with high flow. Additionally, the basilic vein is usually patent due to its location on the posterior aspect of the forearm and the inconvenience of its cannulation for infusions.

In this manuscript, we present a modified UBAVF procedure, which involves placing the patient in the prone position. In the standard supine location, the patient is forced into an intense external rotation (Figure 1A) to prepare the basilic vein, which is often inconvenient, especially in patients with limited elbow motion. As a result, the operator is forced to operate in an uncomfortable position on the medial or posterior side of the forearm. The prone position of the patient makes such a rotation unnecessary (Figure 1B) and allows access to both vessels, i.e., the artery and the vein visible from the top; moreover, in this position, the anatomical conditions are similar to those during the creation of the RCAVF. These two elements can be crucial when it comes to the accuracy of the vascular suture, which may contribute to the success of the procedure. Additionally, even a well-functioning UBAVF is difficult to puncture and less comfortable during haemodialysis, which can result in unintentional haematoma formation. These incidents are likely to have a direct impact on the decreased survival patency of UBAVF compared to classical RCAVF after the initiation of dialysis. The prone position or lateral position of the patient could potentially be used for fistula cannulation. When connected to extracorporeal circulation, the position can be changed to the standard supine position with the assistance of a nurse.

The main limitation of our work is the experience of one case, although in the period from September 2021 to December 2021 we have performed three additional procedures with this modification without any complications.

## 5. Conclusions

The prone position of the patient during the UBAVF procedure provides comfort for the operator as it avoids the lateral rotation of the forearm. It may therefore improve the quality of the arteriovenous anastomosis. Further studies are necessary to test the hypothesis of whether this modification has a positive effect on both patency and the UBAVF function.

## Figures and Tables

**Figure 1 jcm-11-02610-f001:**
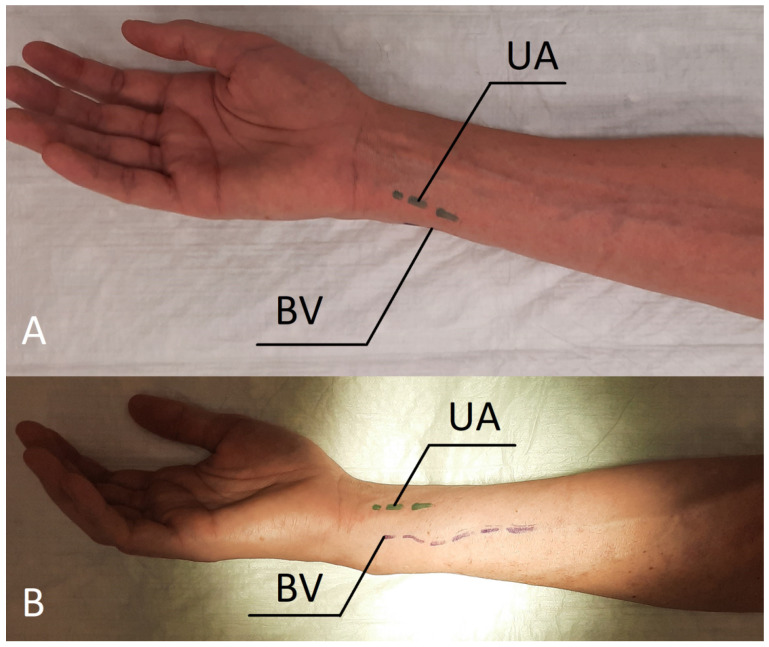
(**A**) Patient in supine position with external rotation of the right forearm, intermittent lines on the skin show the course of the ulnar artery—from this aspect, the basilic vein is not visible. (**B**) Patient in prone position, intermittent lines on the skin show the course of the vessels. Abbreviations: UA—ulnar artery, BV—basilic vein.

**Figure 2 jcm-11-02610-f002:**
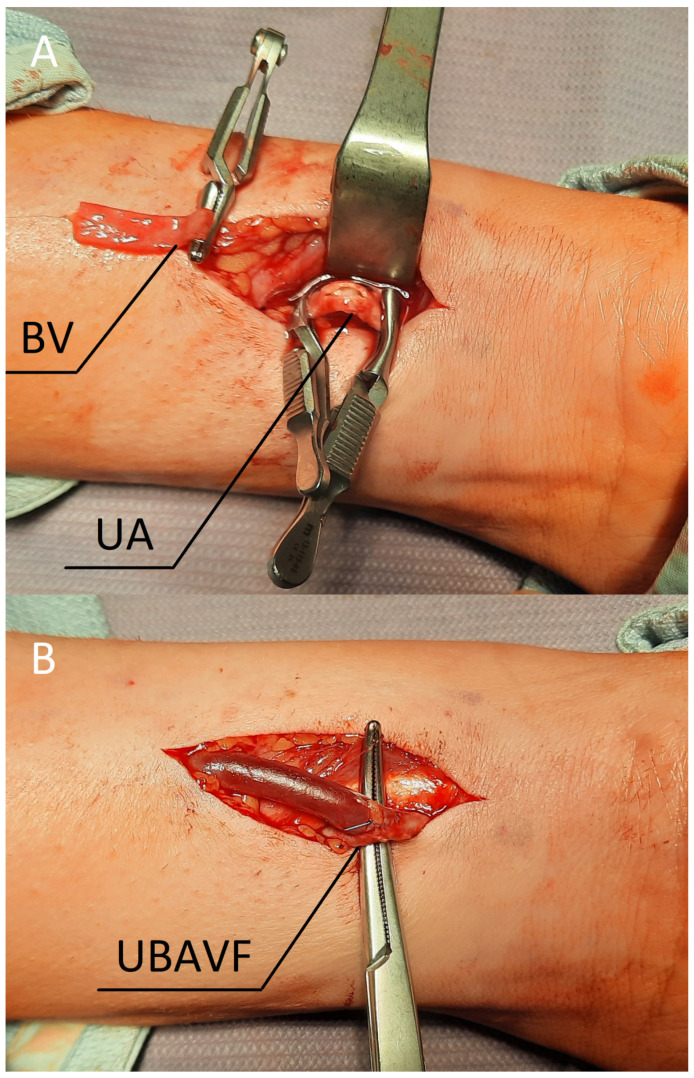
(**A**) Dissected vessels prepared for anastomosis. (**B**) The ulnar-basilic arteriovenous fistula after releasing the clamps. Abbreviations: UA—ulnar artery, BV—basilic vein, UBAVF—ulnar-basilic arteriovenous fistula.

## Data Availability

The data presented in this study are available in this article.

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
