# Peer review of "Prone Position Facilitates Creation of Ulnar-Basilic Arteriovenous Fistula"

_jcm, 2022, doi:10.3390/jcm11092610_

Round 1
Reviewer 1 Report
Please check the attached document.

Author Response
Reviewer 1
In this article, the problems and advantages of conventional UBAVF compared to RCAVF are well summarized and the authors discuss the advantages of new approach towards for creating UBAVF in which the patient is in the prone position during the procedure,for the patient and operator well. One problem is that only one case is presented, and the authors should perform more cases to demonstrate the advantage of this new technique.
Response: Thank you for that opinion. Of course, we agree that one procedure cannot demonstrate the long-term benefits of this modification. In this manuscript, we wanted to focus on the technique of the procedure, so that other physician would have an additional, perhaps more convenient option to perform the usually difficult procedure. UBAVF is a rare type of fistula that is performed when there are no other options. It takes time to collect a larger group of patients, but in the period from September 2021 to December 2021, we performed three additional procedures in this modification without any complications. A prospective study is in our plans.
Reviewer 2 Report
Dear Author,
I agree with the authors hypothesis that placing the patient in the prone position might have easy access to the vessels to form the fistula but the hypothesis that having easy access to the vessels during the procedure might increase the patency and function and viability of the fistula might not be true. Given the position of the AVF , it will be always difficulty to access the UBAVF during the dilaysis sessions and there is always higher chances for the diffuculty accessing the AVF thrice a week, HD needle dislodgement and hematoma/clot formation which might be contributing to the increased failure of the UBAVF compared to the other access that we more commonly use.
And also authors have mentioned that we can encourage lateral and prone position during the HD for the patients with UBAVF which is not an easy option to do practically as we know that patients have to do this approxiamtely 3-4 hours 3 times a week for the many years. Enforcing patients to do this to increase the viability of the UBAVF will have negative impact on the compliance of the HD session by the patients.
Overall, its not appropriate to publish this article explaining about the surgical procedure of the UBAVF and anticipating this will increase the viability of the UBAVF as there more factors like I mentioned above will decide on the success and long term viability of the UBAVF rather than the procedure used during the fistula creation.
It will be more appropriate to do a Prospective study comparing the UBAVF with the other fistulas is needed to assess the long term viabilty.
Author Response
Reviewer 2
Dear Author,
I agree with the authors hypothesis that placing the patient in the prone position might have easy access to the vessels to form the fistula but the hypothesis that having easy access to the vessels during the procedure might increase the patency and function and viability of the fistula might not be true. Given the position of the AVF , it will be always difficulty to access the UBAVF during the dilaysis sessions and there is always higher chances for the diffuculty accessing the AVF thrice a week, HD needle dislodgement and hematoma/clot formation which might be contributing to the increased failure of the UBAVF compared to the other access that we more commonly use.
Response: I am glad that the Reviewer noticed the benefits (better access to vessels) from the UBAVF modification in the prone position. Of course, on the basis of one procedure, we cannot demonstrate its long-term benefits, because, as the Reviewer rightly noted, other factors (difficult puncture, hematoma formation, etc.) may play a role in the functionality of the fistula. In the manuscript, we focus rather on showing an additional option so that the physician who perform these procedures could individually choose a more convenient method of carrying it. In the text, we tried not to form hard conclusions about the influence of this technique on the long-term patency. With this in mind, we have further revised the text to emphasize the aforementioned limitations. We want to emphasize that UBAVF is a rather rare type of vascular access, usually in patients whose other options are exhausted. The text is a case report, but after gathering more patients, we intend to evaluate this technique in terms fistula functionality in a larger group of patients.
And also authors have mentioned that we can encourage lateral and prone position during the HD for the patients with UBAVF which is not an easy option to do practically as we know that patients have to do this approxiamtely 3-4 hours 3 times a week for the many years. Enforcing patients to do this to increase the viability of the UBAVF will have negative impact on the compliance of the HD session by the patients.
Response: Of course, we agree with the Reviewer again that prone position during the 4-hour dialysis procedure may not be accepted by most patients, but we think that the fistula needling itself may take place in this position. When connected to extracorporeal circulation, the position can be changed to the standard supine position with the assistance of a nurse. We have revised this part of the text.
Overall, its not appropriate to publish this article explaining about the surgical procedure of the UBAVF and anticipating this will increase the viability of the UBAVF as there more factors like I mentioned above will decide on the success and long term viability of the UBAVF rather than the procedure used during the fistula creation.
Response: As mentioned above, our intention was not to show long-term viability, but to present a convenient alternative option when creating UBAVF.
It will be more appropriate to do a Prospective study comparing the UBAVF with the other fistulas is needed to assess the long term viabilty.
Response: A prospective study is in our plans.
Reviewer 3 Report
My only opinion on the manuscript is that the author performed a modified version of the patient, so has this new improvement pass medical ethics approval?Has the clinical trial registration already been performed?If not, I think the article should be prudent if it is accepted.
Author Response
Reviewer 3
My only opinion on the manuscript is that the author performed a modified version of the patient, so has this new improvement pass medical ethics approval? Has the clinical trial registration already been performed? If not, I think the article should be prudent if it is accepted.
Response: In our manuscript, we describe a Case report in which we applied modifications to the standard UBAVF fistula. Minor modifications to standard methods do not require the approval of the Biomedical Committee by our hospital. UBAVF is a rare type of fistula and the technique is more difficult than RCAVF due to the access to the vessels, hence the attempt to facilitate the procedure. Before the procedure, the patient received detailed information about what the modification of the procedure is, why we propose it, possible complications and how to proceed in case of complications. In accordance with the guidelines (KDOQI CLINICAL PRACTICE GUIDELINE FOR VASCULAR ACCESS: 2019 UPDATE, https://www.ajkd.org/action/showPdf?pii=S0272-6386%2819%2931137-0) and the idea of "ESKD Life-Plan" patient has the right to decide on the type of treatment method. The patient gave his informed consent. The day before the surgery, he had the opportunity to try this position out. Additionally, a patient experiencing discomfort during the procedure could change his position to the side position, which is considered safe, especially in emergency situations.
Round 2
Reviewer 1 Report
I look forward to the further development of your prospective study.
Reviewer 2 Report
Thank you for making the changes based on my comments.
Reviewer 3 Report
The authors have responded to and revised the opinions of the reviewers.